**Data Availability Statement:** All relevant data are within the manuscript and its Supporting Information files.

**Funding:** There were no funding in the overall accomplishment of this study including the study

# Food taboos and related misperceptions during pregnancy in Mekelle city, Tigray, Northern Ethiopia

**Freweini Gebrearegay Tela** ⬤*, **Lemlem Weldegerima Gebremariam, Selemawit Asfaw Beyene** ⬤

Department of Nutrition and Dietetics, School of Public Health, College of Health Sciences, Mekelle University, Tigray, Ethiopia

* gebrearegayfreweini@gmail.com

## Abstract

### Introduction

Most communities, rural or urban, have taboos regarding foods to avoid during pregnancy, and most have local explanations for why certain foods should be avoided. Such taboos may have health benefits, but they also can have large nutritional and health costs to mothers and fetuses. As such, understanding local pregnancy food taboos is an important public health goal, especially in contexts where food resources are limited. Despite this, information regarding food taboos is limited in Ethiopia. Therefore, this study assessed food taboos, related misconceptions, and associated factors among pregnant women in Northern Ethiopia.

### Methods

A cross-sectional study of 332 pregnant women in antenatal care (ANC) follow-up at selected private clinics in Mekelle city, Tigray, Ethiopa, recruited between April and May, 2017. Using a semi-structured questionnaire, we assessed whether respondents' observed food taboos, what types of foods they avoided, their perceived reasons for avoidance, diversity of respondents' diets during pregnancy, and respondents' socio-demographic characteristics. After reporting frequency statistics for categorical variables and central tendencies (mean and standard deviation) of continuous variables, bivariate and multivariable logistic regression analyses were conducted to identify the socio-demographic factors and diet diversity associated with food taboo practices.

### Results

Around 12% of the pregnant women avoided at least one type of food during their current pregnancy for one or more reasons. These mothers avoided eating items such as yogurt, banana, legumes, honey, and "kollo" (roasted barley and wheat). The most common reasons given for the avoidances were that the foods were (mistakenly) believed to cause: abortion; abdominal cramps in the mother and newborn; prolonged labor; or coating of the fetus's body. Maternal education (diploma and above) (AOR: 4.55, 95% CI: 1.93, 10.31)

design, data collection and analysis, decision to publish, or preparation of the manuscript. We, the authors, made all the necessary efforts from the start to the final manuscript write-up, and no author received a salary from any funder.

**Competing interests:** The authors have declared that no competing interests exist.

and marital status (single) were found to be negatively associated (protective factors) with observances of pregnancy food taboos. Approximately 79% of respondents had pregnancy diets that were insufficiently diverse, although we did not find any statistical evidence that this was associated with adhering to food taboos.

## Conclusion

The misconceptions related to pregnancy food taboos should be discouraged insofar as they may restrict women's consumption of nutritious foods which could support maternal health and healthy fetal development. Health providers should counsel pregnant women and their husbands about appropriate pregnancy nutrition during ANC visits.

## Introduction

Women's nutritional requirements increase during pregnancy, and restrictions on comp-sumption of foods rich in the required nutrients may have negative consequences for a mother as well as for her growing fetus [1]. Undernourished women are more likely to die during pregnancy, to give birth prematurely, and to have babies born prematurely or low birth weight [2]. Newborns who survive infancy but who suffered from fetal growth restriction due to poor maternal nutrition during pregnancy are also at a substantially increased risk of stunting during childhood, and of reduced mental and physical capacity [2].

A healthy maternal diet during pregnancy contains adequate energy, fats, proteins, vitamins and minerals, obtained from consuming a variety of food groups including whole grains, vegetables, fruits, legumes, milk, meat, fish, and nuts [3]. In developing countries like Ethiopia where girls and women usually have inequitable access to healthcare and education, maternal undernutrition remains a major problem. In such countries, insufficient food intake among pregnant women, especially in the 2nd and 3rd trimesters, is common [4]. Most pregnant women don't consume much food during these periods for the fear of having a big baby and a difficult labor.

While adequate dietary intake during pregnancy is affected by many factors including affordability and accessibility, observing food taboos–defined as refraining from eating certain types of foods because of cultural prohibitions–have been commonly reported among pregnant women [5, 6]. Different communities, depending on geography, ecology, religion, tradition, and belief system, have unique dietary profiles and pregnancy food taboos, and communities offers different explanations to consider a given food as taboo [7, 8]. For example, in many cultural contexts, some foods are regarded as taboo due their perceived cause of reproductive health-related problems such as menstrual irregularities or labor- and delivery-related problems. Observing food taboos can also be linked to some factors specific to an individual pregnancy, like how the mother is feeling during her pregnancy [9, 10]. Moreover, consumption of tabooed foods during pregnancy is believed to cause health problems, particularly delay in delivery and or obstructed labor because of big fetus [11].

While such beliefs might originate from the fact that big fetal weight is indeed associated with prolonged and obstructed labor, studies in lower-income countries including Ethiopia have shown that the major cause of obstructed labor is cephalo-pelvic disproportion, which is mainly caused by chronic maternal malnutrition and stunting [12, 13].

Many scholars report the role of religion in encouraging people to observe food taboos [7, 14]. In Ethiopia, Orthodox Christians and Muslims, the two largest religious groups, have

their own food and beverage taboos. To this point, foods are regarded as "halal" (permitted) or "haram" (forbidden) and "yetefekede" (permitted) or "ne'wr" (forbidden) among the Muslim and Christian observers, respectively. For example, eating of animal-based foods during fasting season by observers of Orthodox Christianity–including pregnant women–is considered taboo. Such food taboos have been reported to negatively affect pregnant women and their newborn babies [7, 14, 15].

It has also been observed that who holds household food decision-making power in a given community plays a big role in deciding who should eat what within a given household. To this point, husbands and children are often given priority in the context of food shortage. Though it is debatable whether this rises to the level of a true taboo, it is not acceptable for a mother in a given household to have a particular food before her husband; the husband is given priority and the leftover is taken by the mother and her children [16].

That said, the observance of food taboos among pregnant women can have both negative and positive health consequences. On one hand, food taboos often prevent individuals from consuming foods containing essential nutrients, predisposing them to undernutrition and related morbidity and mortality [8, 10, 17]. Such practices can also cause short- and long-term complications for both mothers and growing fetuses including spontaneous abortion/ miscarriage, fetal growth restriction, preterm delivery, and peri- and post-partum hemorrhage [17, 18]. Subsequently, undernutrition during pregnancy can be a precursor to chronic malnutrition, stunting, and ultimately poor growth and development of the child. Growth restriction in infancy and early childhood is associated with increased risks of obstructed labor and/or giving birth to a low birthweight infant among adolescent girls whose birthweights were low. This can lead to intergenerational cycles of malnutrition and increased risk of developing chronic, non-communicable diseases during adult hood [18]. On the other hand, food taboos are hypothesized to prevent ingestion of food toxins (and related morbidity and mortality) through imposing social costs on pregnant women who eat tabooed foods, with tabooed foods being relatively likely to contain potential pathogens and chemical toxins compared to non-tabooed foods [19, 20]. However, in the context of a population that has experienced multiple generations of under-nutrition, the potential benefits of taboos are likely to be swamped by their nutritional costs, so this study focused on these presumed negative consequences of food taboos.

The observance of food taboos are transmitted socially from one generation to the next, and can become normative in a given community [5, 20]. Information regarding food taboos and other norms in a given community may be transferred from different sources. Grandmothers, elders, and experienced mothers who are considered influential in a given community play central roles in diffusing information regarding which foods are taboo and why, as well as in socially encouraging the subsequent generation to observe taboos [20–22].

Food taboos can be modified within and across generations due to effects of globalization, allowing people to share information outside of local environmental and cultural contexts through different modes, including social media [23]. Furthermore, the role of modern health science in teaching people about the possible adverse effects of avoiding foods containing essential nutrients could also improve awareness and uptake of healthy pregnancy nutrition practices [23].

While many of the specific foods tabooed differ among communities, in most populations, pregnant women commonly refrain from eating certain foodstuffs for different nonscientific and scientific reasons [10, 19, 24, 25]. In Ethiopia, as in many other developing countries [10, 24–26], there are food taboos and misconceptions about the quality and quantity of food pregnant women should or should not eat during pregnancy, affecting their nutritional status [27, 28]. A study in Shashemene district, Ethiopia, indicates that over half (65%) of pregnant women avoided at least one type of food as a result of food taboos, and most frequently avoided food items were: linseed, honey, milk and milk products, meat, egg, fruits, and

vegetables. Such foods were believed: to become coated on the fetal body; to lead to the development of a big baby, causing difficult delivery; to trigger spontaneous abortion/ miscarriage; to indicate evil eye; and/or to cause fetal abnormality [27].

The fact that food taboos negatively affect the dietary intakes of pregnant women underscores the need and importance of assessing food taboos and related misconceptions during pregnancy for the installment of appropriate interventions at local, regional, and national levels. However, the available literatures indicate that little has been done on the issue, and information regarding these practices specifically in Tigray region is lacking. This study, therefore, described the food taboos and related misconceptions during pregnancy in Mekelle city, Tigray region, and assessed the socio-demographic factors that influence their persistence, as well as their possible effects on quality of pregnancy diet.

## Methods and materials

### Study area

The study was conducted in five private clinics found in Mekelle city, the capital of the Tigray region, located in the northern part of Ethiopia, at a distance of 783 Kilometers from Addis Ababa, the capital of Ethiopia. Based on the *Federal Democratic Republic of Ethiopian Central Statistics Agency Report*, the total population of Mekelle city was estimated to be 358,529 in the year 2017 [29]; from this, the estimated number of pregnancies carried to term for the same year was 12,333. The city has one comprehensive specialized hospital, two general hospitals, and nine health centers. The city also has 12 private clinics that provide ANC services.

### Study design and period

A descriptive, cross-sectional study was conducted from April to May 2017.

### Sources and study population

All of the pregnant women attending their antenatal care (ANC) follow-ups from the private clinics in Mekelle city were considered as the target population, and those with ANC follow-up from the randomly selected clinics comprised the study population.

### Inclusion and exclusion criteria

All pregnant women who were permanent residents of Mekelle city, and who were over the age of 20 years at the time of data collection were included. Mothers with hearing and speaking difficulty, and those who were critically ill during the data collection period were excluded from this study.

### Sample size calculation

The sample size was calculated using a single population proportion formula with the assumption of 95% CI and a 0.05 margin of error. Looking at the cross-sectional study done in Hadiya Zone, Northern Ethiopia, 27% of the pregnant women practiced food taboos [28]. Considering P = 27% from that study, the total required sample size was estimated to be 302. We added 10% to accommodate non-responses, and the final target sample size became 332.

### Sampling techniques and procedures

A stratified random sampling technique was used in this study. Five clinics were selected randomly from ANC clinics with the highest attendance rates. The number of mothers to be

included from each clinic was allocated proportionally to clinic size (i.e., number of pregnant women visiting per year). Considering the flow of pregnant women who had been served in the selected clinics during similar months with the data collection period of the previous year as a baseline, number participants recruited from each clinic was calculated as:

<u>Average number of mothers who visit a selected clinic in a month</u> × sample size
Number of mothers who attended in the health facilities over the same month

All pregnant women who came to these clinics for ANC services during the data collection period were taken consecutively until the target sample size (332) was attained.

## Study variables

The variables included in this study were: socio-demographic variables such as maternal age, religion, educational status, marital status, and occupational status; and nutrition-related factors such as food taboos and related misconceptions, meal frequency when non-pregnant, meal frequency when pregnant, Women's Dietary Diversity Score (WDDS), nutrition counseling, and fasting during pregnancy.

## Data collection tools and procedures

A series of closed and open-ended questions were prepared by critically reviewing relevant literature. The open-ended questions were prepared to assess the reasons why pregnant women avoid food items during their pregnancy. Broadly, the questionnaire incorporated questions regarding socio-demographic characteristics, pregnancy-related characteristics, and behavioral factors. Data were collected by trained midwives who work in the selected clinics.

Women's Dietary Diversity Score (WDDS) was calculated from a single 24-hour dietary recall data. All foods and drinks that were consumed the day prior to data collection were categorized into 10 food groups. A score of one was assigned for those who consumed a food item from any of the groups; if not, a score of zero was given. Then, a score out of 10 was computed by summing up the values of all the groups, and it was classified as achieved minimum diet diversity (MDD) ($\geq$ 5) and did not achieve MDD ($<$ 5) [3].

## Data quality assurance

To ensure the quality of the data, a carefully designed data collection tool was prepared. Five data collectors and two supervisors were also trained for two days to develop common understandings of the overall purpose of and data collection procedures used in the study. The questionnaire was pre-tested before the actual data collection on pregnant women attending similar clinics not included in the study, with the test sample reflecting 5% of the target sample. Some modifications to the instrument were made based on the pretest results to ensure the clarity of all questions to both data collectors and respondents.

The questionnaire was translated into Tigrigna (local language), and then back-translated to English by two individuals independently to ensure consistency of concepts. Supervisors followed the data collection process strictly to maximize completeness and quality of the questionnaire responses.

## Data management and analysis

The data were checked and cleaned at the time of data collection and again after data entry. They were then coded and entered into Statistical Package for Social Sciences (SPSS) version 21 for analysis. Categorical variables were summarized as frequencies and percentages. We checked all the continuous variables for normality using Shapiro Wilk's test and they were

found to be appropriately normally distributed (p> 0.05); and these variables were reported as mean ± standard deviation.

Bivariate and multivariable logistic regression analyses were used to assess the associations between observance of food taboos and the independent variables. Biological significance of independent variables was determined using adjusted odds ratios (AOR), 95% Confidence Intervals (CI), and $p$-values. All the predictor variables with a $p$-value of $\leq 0.25$ in the bivariate analyses were included in the multivariable logistic regression model. Variables were analyzed using the enter method, and those with $p$-values $< 0.05$ in the multivariate analysis were identified as statistically significant.

Variance inflation factor (VIF) was used to check for multi-collinearity between the independent variables, and we planned to exclude variables with VIF of $> 5$; however, all variables fell below this threshold, and were thus retained [30]. The model's goodness of fit was checked using Hosmer and Lemeshow test, which indicated that the model fits well ($p$-value = 0.866). In the analysis, the variability of food taboo practice explained by the model was ranged from 6.4–12.5% (Cox and Snell R Square and Nagelkerke R Square).

## Ethical considerations

Ethical approval was obtained from the institutional review board (IRB) of the College of Health Sciences of Mekelle University, and permission was given by the private clinics to proceed with the study. Informed verbal consent was obtained from the study participants, and confidentiality was maintained throughout the study. The participants were well-informed and guaranteed that they had the rights to participate, to refuse or to stop at any time during the data collection process. The procedures of this study constituted a minimal risk to participants, and this was explained to them before the beginning of data collection.

## Result

### Socio-demographic characteristics of the study participants

Almost all (98.8%) of the study participants were from Tigray region, and all of them were residents of Mekelle city. The mean age of the pregnant mothers was 28.5 years (SD ± 3.9). The majority (85.3%) of the participants were followers of Orthodox Christianity. The average family size of the mothers was four. Regarding their educational status, more than half (56.6%) of them had diplomas and above (Table 1).

### Nutrition-related characteristics of the study participants

The staple food consumed by 71% of the participants was injera made of cereals (for example: teff, maize, and sorghum). The majority (91.6%) of them ate three times a day before they became pregnant, and the rest of them ate four or more times a day. Regarding their meal frequency after they got pregnant, 68.1% of them ate four or more times a day. Almost two-thirds (65.1%) of the participants fasted (abstained from consumption of animal-source foods) during pregnancy (Table 2).

Less than one-fourth (20.8%) of the women achieved minimum diet diversity (MDD) (Table 2). Plant-source foods were consumed by a majority of the mothers: 314 (97.6%) of them ate foods made of cereals and roots, 296 (89.2%) ate vitamin A-rich fruits and vegetables, 200 (60.2%) ate dark green leafy vegetables, 312 (94%) ate other fruits and vegetables, and 183 (55.1%) of them ate legumes, nuts and seeds. In contrast, consumption of animal-source foods was not satisfactory among the study participants: 142 (42.8%) of them consumed lean meat, 142 (42.8%) ate fish, 38 (11.4%) ate organ meat, 11 (34%) ate eggs, and 190 (57.2%) ate milk

**Table 1. Socio-demographic characteristics of the study participants (n = 332).**

| Variable | | | Frequency | Percentage |
|---|---|---|---|---|
| Age of the mother | < 30 years | | 198 | 59.6 |
| | ≥ 30 years | | 134 | 40.4 |
| Religion | Orthodox | | 283 | 85.3 |
| | Muslim | | 39 | 14.7 |
| | Others (Catholic and Protestant) | | 10 | 3.0 |
| Marital status | Single | | 46 | 13.9 |
| | Married | | 282 | 84.9 |
| | Widowed | | 4 | 1.2 |
| Educational level | Secondary education or below | | 144 | 43.4 |
| | Diploma and above | | 188 | 56.6 |
| Occupational status of the mother | | | | |
| | Student | | 16 | 4.8 |
| | Housewife | | 112 | 33.7 |
| | Government employee | | 78 | 23.5 |
| | Non-governmental employee | | 45 | 13.6 |
| | Self–employed | | 81 | 24.4 |
| Ownership of monetary resources | | Father | 130 | 39.2 |
| | | Mother | 23 | 6.9 |
| | | Both jointly | 179 | 53.9 |
| Family size | < 4 individuals | | 179 | 53.9 |
| | ≥ 4 individuals | | 153 | 46.1 |
| Parity (Number of childbirths) | Primipara (1 childbirth) | | 117 | 51.8 |
| | Multipara (1–4 childbirths) | | 100 | 44.2 |
| | Grand multipara (≥5 childbirths) | | 9 | 4.0 |

and milk products. Out of the total participants, 31 (11.8%) of them reported observing one or more food taboos and had low MDDS; and 7 (10.8%) of them reported not observing food taboos and nonetheless had low MDDS.

**Table 2. Nutrition related characteristics of the participants (n = 332).**

| Variable | Category | Frequency | Percentage |
|---|---|---|---|
| Meal frequency when non-pregnant | 3 times | 304 | 91.6 |
| | ≥ 4 times | 28 | 8.4 |
| Meal frequency when pregnant | 3 times | 106 | 31.9 |
| | ≥ 4 times | 226 | 68.1 |
| Nutrition counseling received | Yes | 86 | 25.9 |
| | No | 246 | 74.1 |
| Fasting during pregnancy | Yes | 216 | 65.1 |
| | No | 116 | 34.9 |
| WDDS | Achieved MDD | 69 | 20.8 |
| | Didn't achieve MDD | 263 | 79.2 |
| Practice of food taboo & MDDS | Didn't achieve MDD | 31 | 11.8 |
| No food taboo practice & MDDS | Didn't achieve MDD | 7 | 10.8 |

WDDS: Women's dietary diversity score; MDD: minimum dietary diversity: mothers who consumed at least five out of the ten food groups

## Food taboos and related misconceptions during pregnancy

Thirty-eight (11.5%) (95 CI: 7.8, 15.1) of the participants avoided at least one type of food during their current pregnancy for different reasons. Legumes were reported as taboo foods by 45.7% of the pregnant mothers who observed food taboos. Furthermore, 22%, 15.8%, 13.2%, 10.5%, 8.1%, and 8% of the pregnant women avoided mustard, porridge, bananas, whole grains in the form of "kollo", honey, and milk products (yogurt and milk), respectively.

The most common reasons given for avoiding legumes (beans and chickpeas) were that they are believed to cause abdominal cramps in both mother and fetus, to prolong labor, to exacerbate labor pain, and to cause abortion. Similarly, whole grains in the form of "kollo" were believed to exacerbate labor pain, and to cause postpartum abdominal cramps, heartburn, and nausea in the mother. Porridge, bananas, and milk products were also avoided by some mothers because of the perception that they become coated to the body of the fetus and make the baby very big, causing difficult/prolonged labor. Honey was considered taboo by some respondents because of the perception that it causes abortion, and exacerbates labor pain. Mustard was also perceived by some mothers to cause abortion (Table 3).

## Socio-demographic factors associated with observing food taboos during pregnancy

In bivariate logistic regression: maternal age, maternal education, fasting during pregnancy, maternal occupation, marital status, and religion were found to be statistically associated with likelihood of observing food taboos at $p$-values <0.25; these variables were included in the multivariate analysis.

In the multivariate analysis: maternal education and marital status were found to be negatively associated (i.e., were protective factors against) with food taboo observance during pregnancy at $p$-value <0.05 (Table 4). The odds of observing food taboos was 4.6 times higher among women who had not attended any tertiary education (AOR: 4.55, 95% CI: 1.93, 10.31) when compared to those who held diplomas (or higher credentials). The odds of observing food taboos was 0.22 times *lower* among single and widowed mothers when compared to married women (AOR: 0.22, 95% CI: 0.05, 0.97).

## Discussion

This study was aimed at assessing food taboos and related misconceptions during pregnancy in Mekelle city, Tigray, northern Ethiopia. Like other regions of the country [27, 28], food

**Table 3. Foods study participants avoided during pregnancy, and reasons given for their avoidance (n = 38).**

| Type of food taboo | Frequency (%) | Reasons for the avoidance of these food items |
|---|---|---|
| Legumes (beans, and chickpea) | 16 (45.5%) | Causes abdominal cramps and prolongs labor, and exacerbates labor pain, and abdominal cramps to the fetus, and causes abortion |
| Mustard | 8 (22%) | Causes abortion or abdominal cramps in the newborn |
| Porridge | 6 (15.8%) | Coated to the body of the fetus and makes the baby very big causing difficult labor |
| Banana | 5 (13.2%) | Coated to the body of the fetus, makes the baby very big causing difficult labor |
| "Kollo" (roasted wheat and barley) | 4 (10.5%) | Causes postpartum abdominal cramps, heartburn, nausea, and exacerbates labor pain |
| Honey | 3 (8.1%) | Causes abortion or exacerbates labor pain |
| Yogurt and milk | 3 (8.1%) | Coated to the body of the fetus, and makes the baby very big causing difficult labor |

**Table 4. Socio-demographic factors associated with food taboo practice among pregnant women in Mekelle city, Tigray, Ethiopia, 2017.**

| Variables | Categories | Food taboo practice | | COR (95% CI) | AOR (95% CI) |
|---|---|---|---|---|---|
| | | No | Yes | | |
| Age | <30 years | 173 | 25 | 1.35 (0.66, 2.73) | 1.27 (0.61, 2.64) |
| | ≥30 years | 121 | 13 | 1 | 1 |
| Religion | Orthodox | 249 | 34 | 1.54 (0.52, 4.54) | 1.92 (0.60, 6.13) |
| | Muslim and protestant | 45 | 4 | 1 | 1 |
| Marital status | Single | 44 | 6 | 0.29 (0.07, 0.22) | 0.22 (0.05, 0.97)* |
| | Married | 246 | 36 | 1 | 1 |
| Educational level | Secondary education or below | 117 | 27 | 3.71 (1.77, 7.78) | 4.55 (1.93, 10.31)*** |
| | Diploma and above | 117 | 11 | 1 | |
| Occupational status of the mother | Housewife | 111 | 24 | 0.64 (0.27, 1.51) | 1.10 (0.39, 3.04) |
| | Government and non-governmental employee | 112 | 11 | 0.69 (0.32, 1.51) | 0.86 (0.36, 2.01) |
| | Self-employee | 71 | 10 | 1 | 1 |
| Fasting during pregnancy | Yes | 101 | 15 | 0.80 (0.40, 1.60) | 0.85 (0.40, 1.81) |
| | No | 193 | 23 | 1 | 1 |

*p-value <0.05

***p–value <0.0001

taboos and related misconceptions influence the dietary practice of some pregnant women in the study area; specifically, 11.5% (95 CI: 7.8, 15.1) of the participants avoided at least one type of food during their current pregnancy because they believed the food(s) to be taboo and that eating those foods would pose health risks to themselves or the fetus. The prevalence of food taboos observed in this study appears to be relatively low compared to studies conducted elsewhere in Africa in general and Ethiopia in particular [27–32]. This lower prevalence might be because this study was conducted in private clinics, and the participants have relatively higher educational levels, with more than half of participants holding diplomas or higher-level credentials. The fact that all of the study participants in this study are urban (Mekelle city) residents might also improve their access to information and awareness of appropriate nutrition during pregnancy.

The common food items avoided by those pregnant women who followed food taboos in this study were legumes, porridge, banana, honey, mustard, and whole grains in the form of "kollo", with reasons for the taboos differing by food type. This finding is consistent with literature on pregnancy food taboos from a variety of other social and ecological contexts, which also shows that many pregnant women refrain from eating a variety of nutritious food items during pregnancy due to socially-transmitted beliefs (often misconceptions) that these foods may harm their pregnancies. A study conducted among rural women of Surendranagar district in India showed, for example, that women commonly avoid fruits such as papaya, ground nuts and citrus foods because of the perception that they can cause abortion, placental disruption and difficult labor [31]. A study among rural women of Aligarh, also in India, similarly revealed that most of the pregnant women in the community avoid papaya, fish, citrus foods, and ground nuts because they are believed to cause abortion, placental abruption, itching, and seizure [32].

It is obvious that observance of food taboos and adhering to related misconceptions about dietary prohibitions can negatively affect the nutrition and health status of pregnant women as well as the health, development, and life-long wellbeing of their growing babies [2, 33, 34]. In this study, some pregnant mothers were prohibited from consuming food items such as whole grains in the form of "kollo", and legumes such as beans and chickpeas because they were

believed to cause abdominal cramps during labor, to prolong labor, and to cause abdominal cramps to the newborn. Avoidance of such whole grains and legumes may negatively affect the dietary intakes of these women, whereas dietary diversity recommendations for pregnant women emphasize the need for pregnant women to eat diverse foods with adequate energy, protein, fat, fiber, and micronutrients [3]. Notably, whole grains, wheat bran, and other high fiber foods are used to relieve constipation, a common source of discomfort in pregnancy, and legumes can provide high quality protein when they are complemented by whole grain-containing foods.

Pregnant women also refrained from eating animal-source foods such as milk and yogurt, which are rich in high quality (complete) proteins [25]. This was similar to a study conducted in Hadiya Zone, Ethiopia, which demonstrated that pregnant women were restricted from eating milk and cheese for fear of difficult labor and delivery [28]. A similar study in Sudan also reported that 41.5% of the pregnant women refrained from drinking milk [35]. This might lead to poor pregnancy weight gain, and increase the risk of giving birth to a low birth weight baby. Moreover, low consumption of animal-source foods during pregnancy could also deprive pregnant women of various essential nutrients, leading to protein, energy, and micronutrient deficiencies, particularly insofar as many protein-rich foods are also good sources of calcium, iron, and vitamin B-complex [36].

Bananas were also considered a taboo food by some of the study participants, who believed that banana can become coated to the body of the fetus, leading to the development of big babies (fetal macrosomia), causing difficult labor. This finding was in line with a study in Shashemene, Ethiopia, which revealed that pregnant women were restricted from eating fruits [27]. Avoiding eating bananas during pregnancy may be problematic because bananas are not only cheap and energy-dense, but are also rich in potassium, which can improve the health of the heart of the mother as well as her growing fetus. Bananas also contain many other micronutrients such as vitamin $B_6$ and minerals, as well as fiber; they also have antioxidant properties [37].

Pregnant women were also restricted from eating honey because it was believed by some to cause abortion and exacerbate labor pain. This finding aligned with a previous Ethiopian study, conducted in the Shashemene region, which also reported that some pregnant women avoided eating honey because of the perception that it may be an abortificant [27]. The scientific evidence, in contrast, indicates that honey poses no known health risks to pregnant women or fetuses, and contains high amounts of carbohydrates, proteins, minerals, and multiple antioxidants [38].

This study showed a significant association between observance of food taboos by pregnant women and their level of education (AOR: 4.55, 95% CI: 1.93, 10.31). This can be condensed to: This finding aligned with at least three previous studies carried out in East African populations, including one in Sudan, one in Shashemene District, Ethiopia, and one in both Nigeria and Sudan. Results of all studies, like this one, indicated that higher maternal education was associated with a reduced likelihood of observing pregnancy food taboos [27, 31, 39] This might be due to the knowledge that they gained from formal education and from reading which may simultaneously boost their healthy eating practice.

Marital status was also found to be negatively associated with food taboo practice among pregnant women (AOR: 0.22, 95% CI: 0.05, 0.97). Single and widowed mothers were less likely to observe food taboos than married women. This might be due to the fact that, particularly in developing countries like Ethiopia, women mostly abide by and respect the ideas and/or beliefs of their husbands. This was in line with the finding from a study in Ghana which showed that respect for parents was a motivating factor for avoiding eating prohibited foods during pregnancy [26]. A review of evidence on traditional beliefs and practices from Asian countries also

suggests that nutrition education should not only be provided to mothers but also to husbands and parents [40]. In the bivariate analysis, the number of pregnant mothers who reported observing food taboos and who had low MDDS was higher than mothers who did practice food taboo and had low MDDS though the difference was not statistically significant. Avoiding nutrient-rich foods during pregnancy is very likely to affect the mother's dietary quality, so this lack of statistical evidence for an association might be due to insufficient statistical power for this particular *post hoc* analysis (not planned during our initial power calculation). Further research with a bigger sample size to explore the association between food taboos and MDDS will be conducted in future work.

A strength of this study is that recall bias was limited because the pregnancy-related information was collected at the time of pregnancy. However, its cross-sectional nature may also be considered a limitation of this study. Future work will thus investigate the effect of food taboos on maternal nutritional status and on pregnancy outcomes longitudinally, in a cohort study. Another shortcoming of the study was its recruitment focus on private ANC clinics from an urban-only context, as this may have over-represented the responses of well-educated women. Including participants from rural communities would improve the generalizability of the results. Furthermore, we did not collect data on the other forms of food avoidances (visceral aversions, nausea, disgust, taboos not specific to pregnancy; avoidances due to allergies or sensitivities and/or family preferences), and understanding these different potential causes of food avoidances would have allowed us to make stronger claims regarding how to intervene and to help pregnant women eat nutritiously.

As a conclusion, adherence to culturally-based food beliefs is evident in pregnant women found in Mekelle city. Educational status and marital status were found to be negatively associated with observing food taboos during pregnancy. Thus, there is a need for nutrition education and awareness creation about the presumed nutritional consequences of following food taboos. As a short-term intervention, this kind of education should be developed and disseminated during ANC follow-ups, and should target not only pregnant women but also their husbands. In the longer-run, the literacy level of mothers should be improved across the life cycle, from early childhood through adolescence.

## Supporting information

**S1 File. Questionnaire for data collection.**
(DOCX)

**S2 File. Flow chart for population proportional allocation.**
(DOCX)

**S3 File. Data set of the collected data.**
(SAV)

## Acknowledgments

The authors would like to acknowledge all of the respondents for their information, and we also thank the administrative bodies of the clinics for allowing as to collect this information from their institutions.

## Author Contributions

**Conceptualization:** Freweini Gebrearegay Tela, Lemlem Weldegerima Gebremariam, Selemawit Asfaw Beyene.

**Data curation:** Freweini Gebrearegay Tela, Lemlem Weldegerima Gebremariam, Selemawit Asfaw Beyene.

**Formal analysis:** Freweini Gebrearegay Tela, Lemlem Weldegerima Gebremariam, Selemawit Asfaw Beyene.

**Funding acquisition:** Freweini Gebrearegay Tela, Lemlem Weldegerima Gebremariam, Selemawit Asfaw Beyene.

**Investigation:** Freweini Gebrearegay Tela, Lemlem Weldegerima Gebremariam, Selemawit Asfaw Beyene.

**Methodology:** Freweini Gebrearegay Tela, Lemlem Weldegerima Gebremariam, Selemawit Asfaw Beyene.

**Project administration:** Freweini Gebrearegay Tela, Lemlem Weldegerima Gebremariam, Selemawit Asfaw Beyene.

**Resources:** Freweini Gebrearegay Tela, Lemlem Weldegerima Gebremariam, Selemawit Asfaw Beyene.

**Software:** Freweini Gebrearegay Tela, Lemlem Weldegerima Gebremariam, Selemawit Asfaw Beyene.

**Supervision:** Freweini Gebrearegay Tela, Lemlem Weldegerima Gebremariam, Selemawit Asfaw Beyene.

**Validation:** Freweini Gebrearegay Tela, Lemlem Weldegerima Gebremariam, Selemawit Asfaw Beyene.

**Visualization:** Freweini Gebrearegay Tela, Lemlem Weldegerima Gebremariam, Selemawit Asfaw Beyene.

**Writing – original draft:** Freweini Gebrearegay Tela, Lemlem Weldegerima Gebremariam, Selemawit Asfaw Beyene.

**Writing – review & editing:** Freweini Gebrearegay Tela, Lemlem Weldegerima Gebremariam, Selemawit Asfaw Beyene.

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
