## [Decision Letter · Decision Letter 0]

24 Mar 2020

PONE-D-20-06088

Food taboos and related misperceptions during pregnancy in Mekelle city, Tigray, Northern Ethiopia

PLOS ONE

Dear Mrs. Tela,

Thank you for submitting your manuscript to PLOS ONE. After careful consideration, we feel that it has merit but does not fully meet PLOS ONE’s publication criteria as it currently stands. Therefore, we invite you to submit a revised version of the manuscript that addresses ALL the points raised during the review process.

We would appreciate receiving your revised manuscript by May 08 2020 11:59PM. To enhance the reproducibility of your results, we recommend that if applicable you deposit your laboratory protocols in protocols.io, where a protocol can be assigned its own identifier (DOI) such that it can be cited independently in the future. For instructions see: http://journals.plos.org/plosone/s/submission-guidelines#loc-laboratory-protocols

We look forward to receiving your revised manuscript.

Kind regards,

Frank T. Spradley

Academic Editor

PLOS ONE

Journal Requirements:

2. Please include additional information regarding the survey or questionnaire used in the study and ensure that you have provided sufficient details that others could replicate the analyses. For instance, please include a copy of the questionnaire also in the original language  as Supporting Information. Moreover, please include more details on how the questionnaire was originated and  pre-tested, and whether it was validated.

"no funding"

a)    Please provide an amended Funding Statement that declares *all* the funding or sources of support received during this specific study (whether external or internal to your organization) as detailed online in our guide for authors at http://journals.plos.org/plosone/s/submit-now.  

b)    Please state what role the funders took in the study.  If any authors received a salary from any of your funders, please state which authors and which funder. If the funders had no role, please state: "The funders had no role in study design, data collection and analysis, decision to publish, or preparation of the manuscript."

Reviewers' comments:

Reviewer's Responses to Questions

**Comments to the Author**

1. Is the manuscript technically sound, and do the data support the conclusions?

Reviewer #1: Partly

Reviewer #2: Partly

2. Has the statistical analysis been performed appropriately and rigorously? 

Reviewer #1: Yes

Reviewer #2: Yes

3. Have the authors made all data underlying the findings in their manuscript fully available?

Reviewer #1: Yes

Reviewer #2: Yes

4. Is the manuscript presented in an intelligible fashion and written in standard English?

Reviewer #1: No

Reviewer #2: Yes

5. Review Comments to the Author

Reviewer #1: “Food taboos and related misperceptions during pregnancy in Mekelle city, Tigray, Northern Ethiopia” reports new data on food avoidances during pregnancy in a sample of 332 pregnant people from a large urban centre in Ethiopa. The data themselves are fascinating and this paper has the potential to contribute to the literature on the impacts of food avoidances on diet during pregnancy. Given that maternal diet, health, and wellbeing from just before conception through pregnancy have outsized effects on subsequent health and wellbeing for both mothers and children, the data have obvious implications for public health. Additionally, as selection is relatively strong during the earliest stages of life, the data also may be of relevance to understanding a key facet of how human biocultural evolution operates in contemporary urban Ethiopia.

However, I cannot recommend the publication of this manuscript in its current form, especially in light of the fact that PLoS ONE places NO restrictions on manuscript length. The authors do almost nothing to situate the data reported within a larger academic debate, and do not outline any theoretical perspective, specific hypotheses, or predictions. The most obvious perspective to bring to this manuscript is the Developmental Origins of Health and Disease (DOHaD) framework (which is sort of but not satisfactorily alluded to), although biocultural theory, dual inheritance theory, human behaviour ecology, evolutionary psychology, or some kind of network theory could also be compelling. Alternatively, an in-depth ethnographic framing describing the historical particulars (i.e., the social as well as dietary significance of the foodstuff in the city and its sub-cultures) underpinning each of the common taboos would be interesting. Some intellectual framework needs to be established and justified for this to be publishable.

In addition to this overarching issue, a few other important points:

1) The authors seemingly do not disentangle pregnancy food taboos from other kinds of food avoidances (visceral aversions, nausea, disgust; taboos not specific to pregnancy; avoidances due to public health guidelines; avoidances due to allergies or sensitivities or family preferences). It’s not clear whether data were collected on whether pregnant people would usually eat a certain food and even wanted to eat it but didn’t because of a taboo or didn’t because it made them nauseous and there also happened to be a taboo about it. This issue needs to be better outlined in the methods; if data are available on other reasons for food avoidances, they should be reported and discussed; if data are not available, this should be discussed as a limitation and avenue for future research.

2) No alternatives to the assumption that the taboos are problematic were presented. Yes, the pregnant people are missing out on nutrients, but are they also avoiding pathogens, or preventing fetal overgrowth, or guaranteeing that nutrients go to support other household members? I would expect to see some thoughtful discussion and familiarity with the literature around these points.

3) The authors don’t seem to look at the extent to which pregnant people are perhaps compensating for nutrient losses perhaps through experiencing cravings or following other dietary recommendations and thus filling in some of the nutrient gaps created by food avoidances. My colleagues and I found that there was evidence of pregnant people in Fiji eating other foods with similar nutrient profiles to avoided foods (McKerracher et al. 2016; see also Henrich and Henrich 2010)

And a couple of minor ones:

1) Education is a major predictor of not adhering to food taboos, and the public health recommendation the authors make then is to improve antenatal education (e.g. lines 271-272). I’d suggest going to pre-conception and even to adolescence to really make a big difference. See the work by MacNab and Mukisa in South Africa, by Jackie Bay and colleagues in New Zealand and the Cook Islands, by Mary Barker/ Kathy Townsend and colleagues with the Each-B trial in Southampton UK, and so on. Also check out the Lancet review on preconception interventions (Stephenson et al. 2018).

2) I’d generally like to see a little more context on the overall health of the pregnant population in Merkell in general and in the sample in particular. What are infant and maternal mortality like? Are there high rates of GDM? High rates of pre-eclampsia/ hypertension? High rates of fetal growth restriction? Pre-term birth? Average completed family size? Food security?

3) Please unpack the dietary diversity score. This should be clearly outlined in the main text.

Lastly, the manuscript would benefit from a bit more spell and grammar checking. Hopefully the editors can provide this or at least some financial assistance as the team is from a lower-income country. I’d also be happy to help with proofing the revision if I was sent a word doc or google doc file.

To the authors: Cool data and lots of potential here! Just needs some more theory and context. Also, I apologize for not providing as much specific feedback or as much positive feedback as I usual do when reviewing. I am really short on time for this review because of, well, the global pandemic and needing to look after my kids who no longer have school/ day care while all of Canada is in a public health lockdown.

Kindest regards,

Luseadra McKerracher

(intentionally signed here, but don't necessarily want my name published as a reviewer - just think it's a nice courtesy to the authors)

Reviewer #2: Remarks to the Manuscript by Tela FG et al. on

„Food taboos and related misperceptions during pregnancy in Mekelle city, Tigray, Northern Ethiopia“

We strongly agree that the level and impacts of food taboos on the health of pregnant mothers, fetus and pregnancy outcome need to be explored. To reduce its related negative impacts, sound and appropriate evidence informed interventions need to be implemented. In this sense, we appreciate the researchers for investigating and bringing scientific information on food taboos and related misconceptions among pregnant mothers of Mekelle city.

Abstract:

Ln 27: Add “in” after the word “limited” and before the word “Ethiopia”.

Ln 30-31: the timeline should be specified as “1st of January to 30th of June 2017”.

• However, we wonder why this timeline of data collection is different from what is mentioned on Ln 87, where it was stated that “The study was carried out from April to May 2017….”. Which one is correct?

Ln 33/34: please, do the following changes: “described” to “presented”, “frequency” to “frequencies” and “percentage to “percentages”. Check also Ln144-145.

Ln 43: you may add “Ethiopia” as a keyword.

Introduction

Ln 55-59: The sentence is very long and it needs to be broken-down in two short and informative sentences.

Ln 56: “under-nutrition” should be written without hyphen as “undernutrition”

Ln 60-63: please, cite the references because when you say “…have been reported…”, you are referring to previous research facts, not of yours.

Ln 67: add comma (,) after the word, “especially”, and why is it “especial”? it could be a good idea if you can paraphrase this sentence.

Ln 76: the word “closely” is a very ambiguous expression and you may need to replace it by “negatively” or any other clear word, which can clearly indicate the nature &/ or direction of the association.

Ln 79: avoid the semicolon (;) and better replace it by comma (,).

Methods

Ln 83: The “Study period” sub-section could be mentioned together with the “Study period” under one sub-topic, written as “Study design and period”. Then, the section “Study design and period” should come after the “Study area”.

• Under the “Study design and period”, the study objective must not be mentioned here. Rather, you may paraphrase as in the following: “A descriptive cross-sectional study was conducted from April to May 2017 (make sure to put the correct timeline).”

Ln 87: The study period referred here is different from the one mentioned in Ln 30-31. Timing is a very important dimension in epidemiologic studies. Why is it different? Which one is correct?

Ln 89-92: What is the source of the information? Please, cite the source. What does “2016/2017” mean? Does it refer for either of the years or for both?

Ln 91-92: The sentence is unclear and it may mean that all these health stations are there only to provide ANC services. Please, paraphrase it.

Ln 101: for your future research plan, you need to know three important points regarding sample size calculation. First, if your calculated sample size has a decimal, the sample size must be ROUNDED UP regardless of the value of the decimal. Many researchers do not do this, but from a statistical point of view; this is what has to be done. For instance, in your case, the calculated sample size it 332.2 and it has to be approximated up and it should be 333. Second, whenever you are considering “none response rate”, you should calculate it using the formula (sample size divided by the response rate in decimal) and it should not be added just by taking 10% of the calculated sample size. In your case, the initial actual sample size is 302.9 (≈303) and taking a 10% non-response rate (303/0.90= 33.7≈ 34, and the final sample size (at its best scenario) would have been 337. Third, if your source population (in this case the total number of pregnant women in the study area) is small (<10,000), the sample size need to take in to account, a population correction factor. Based on your report for the year prior to your study, your study population is not a small population (check Ln 91).

Ln 97: You may replace “Eligibility criteria” by “Inclusion and exclusion criteria”.

Ln 107: The sub-topic “sampling techniques” could be modified as “Sampling techniques and procedures”. What sampling technique was used? It is always important to mention the sampling method. From the texts, it seems that you have employed a stratified random sampling technique. You may address the following major concerns.

• Ln 108: Because your study clinics were randomly selected from the institutions with a “better flow of pregnant women for ANC services”, what does this imply to your findings? There could be the possibility of “selection bias” because participants of your study were from clinics, which had a “better flow” of pregnant women for ANC services. The profile (socio-demographic and economic attributes) of the participants of your study may differ from those who attended other health institutions, which did not have a “better flow”, and the prevalence of food taboo and related misconceptions could be different. You may discuss it.

• Ln 108: it may be better to say “ANC clinics with a higher attendance rate”.

• Ln 109 and Ln 112: If the allocation to each of the five health institutions was proportional. This implies the total number of pregnant women was known for each of these study clinics. How was the proportional allocation made? How was the sampling interval defined (N/n)? For a clarity purpose, you may submit a “supplementary” table or flowchart that shows the proportional allocation and its respective sampling interval.

Ln 120: you may modify the sub-topic to “Data collection tools and procedures”.

• Ln 122: “ in the way that they can address” may be replaced by “to asses “. You should replace “literatures” by “literature”.

• Ln 125: Your results show no behavioral factors. So, you may make it clear.

• Ln 132: contractions, like “didn’t”, should be avoided in scientific writing. Replace “didn’t” by “did not”.

Ln 135: Please, mention the number of data collectors and supervisors.

Ln 136: You may replace “overall purpose and methodology” by “overall purpose and data collection procedure of the study” if this makes sense to you.

Ln 137: On how many women and where was the pilot test implemented? Who were the actual participants of the pilot-test study?

Ln 137-139: It is great that you did the translations and retranslations. Who did both translations to the local language and the retranslation back to English? Was it done by the same individual/s? You may briefly explain these points.

Ln 145: add comma after “Finally”.

Ln 145: replace “cross-tabulations” by “Chi-square”. As shown in your table (check Ln 202), you have zero cell counts. In this case, you should use Fishers Exact Test as an alternative to chi square test. You should write the statistical parameter you used to summarize for the continuous variables.

• We strongly suggest using appropriate statistical analysis to identify the factors associated with the food taboo, which has a prevalence of 11.5%. Authors should use the data to its maximum potential and may have ethical aspect from a statistical point of view. We suggest to the authors to address two important points (based on their result from Table 4). First, they should combine the levels categories of some independent in to broader categories to avoid the problem of convergence. Second, they should test for presence of multicollinearity, which could bias their measure of association like the odds ratio or prevalence ratio. This is not a must to do, but we encourage you to do it.

• If the authors decide to run a model to identify the factors associated with food taboo, the objective and methods part of the study need to be modified accordingly.

Results

• General comment: use present tense when referring to tables, figures and graphs. Otherwise, you must use simple past tense to report your results.

• Avoid using the word “significant”. Nowadays, it is highly encouraged using other terms like “increased or decreased” if your finding is statistically significant.

Ln 160: Age of participants was reported in mean and standard deviation. Was it normally distributed? If not, you need to report the median and interquartile range. This works for all continuous variables you have (like family size).

Ln 162: replace “were diploma and above” by “had diploma and above”.

Ln 169-171: Check subject-tense agreement (simple past tense should be used). Replace “eat” to “ate”, and “get” by “got”.

Ln 185: avoid the comma (,) after the word “beans”

Ln 191: avoid “also” which is mentioned after “Honey was” because you have used it in the next sentence.

Ln 198-199: replace “are” by “were”.

Ln 180: report the 95% confidence interval of the prevalence.

Ln 198-199: replace “are” by “were”.

Ln 202: please, make sure that you reported Fishers Exact Test for the cell counts with zero values or the expected value is less than 5.

Discussion

• Generally, your findings are well discussed, but if you need to run additional statistical analysis, the discussion section will certainly need additional interpretations based on the new outputs.

Ln 249-253. This lacks critical interpretation of the results to create awareness on the potential harms of consuming honey on the health of their newborns. You should also critically discuss the various negative aspects of honey use. You need to cite more relevant articles in this field, which can help you to improve the respective discussion.

• Please, cite more important articles, you should at least cite the work of Ajibola et al. (doi: 10.1186/1743-7075-9-61)

• Natural honey can be contaminated by c.botulinum, which is fatal, and definitely, it should not be given to children less than 1 year. Check CDC’s official webpage (https://www.cdc.gov/botulism/)

6. PLOS authors have the option to publish the peer review history of their article (what does this mean?). If published, this will include your full peer review and any attached files.

Reviewer #1: No

Reviewer #2: Yes: Semaw Ferede Abera

---

## [Author Response · Author response to Decision Letter 0]

5 May 2020

Response to reviewers

Comments raised by the academic editor:

1. Please ensure that your manuscript meets PLOS ONE' style requirements, including those for file naming.

Response: - Thank you so much for your credible concern dear! I have read the PLOS ONE' guideline and revised the manuscript as much as possible. 

2. Please include additional information regarding the survey or questionnaire used in the study and insure that you have provided sufficient details that others could replicate the analysis. For instance, please include a copy of the questionnaire also in the original language as supporting information. Moreover, please include more details on how the questionnaire was originated and pre-tested, and whether it was validated.

Response: - Thank you for your valid comment dear! I had wrongly attached inappropriate draft of the questionnaire; but now I have attached the right one in its English and Tigrigna (the local language) version as a supporting information. The questionnaire was developed by reviewing different literature. To be honest, the questionnaire was not validated; as there were some studies conducted in Ethiopia that used similar data collection tool, we simply reviewed these literature and developed the questionnaire. 

3. We not that you have indicated that data from this study are available up on request. Plose only allows data to be available up on request if there are legal or ethical restrictions on sharing data publicly. In your revised cover letter, please address the following prompts:

a. If there are ethical or legal restrictions on sharing a de-identified data set, please explain them in detail (example: data contain potentially identifying or sensitive patient information) and who has imposed them (an ethics committee). Please also provide contact information for a data access committee, ethics committee or other institutional body to which data requests may be sent.

b. If there are no restrictions, please upload the minimal anonymized data set necessary to replicate your study finding as either supporting information files or to a stable, public repository and provide us with the relevant URLs, DOIs, or accession numbers. 

Response: Sorry for the misinformation I made on behalf of all the authors dear! It was wrongly stated that data are available up on request. The data in this study have no ethical or legal restrictions on sharing publicly, and all data supporting the results of this study are made available in the manuscript and as supporting information. If I have forgotten anything important, I can provide it. I have incorporated this idea in the revised cover letter. 

4. Thank you stating the following financial disclosure ‘’no funding’’

a. Please provide an amended funding statement that declares *all* the funding or sources of support received during this specific study (whether external or internal to your organization) as detailed online for authors at http://journals.plos.org/plosone/s/submit-now

Response: Thank you for noticing this important issue dear! There is no any funding we received for this work. Furthermore, as the Plose guideline do not allow to incorporate funding statement in the manuscript, I have removed it up on revision of the manuscript.

b. Please state what role the funders took in the study. If any authors received a salary from any of your funders. Please state which authors and which funder. If the funders had no role, please state: ‘’the funders had no role in study design, data collection and analysis, decision to publish, or preparation of the manuscript.’’

• Please include your amended statements with in your cover letter; we will change the online submission form on your behalf.

Response: Thank you for your constructive comment dear! I will like to clarify that there were no funding in the overall accomplishment of this study including the study design, data collection and analysis, decision to publish, or preparation of the manuscript. We, the authors, made all the necessary efforts from the start to the final manuscript write-up, and no author received a salary from any funder. I have incorporated this idea in the revised cover letter.

Reviewers’ comments:

Comments raised by reviewer 1:

Thank you dear for your encouraging words and your relentless effort to boost the quality of our manuscript. All of your comments and suggestions are really appreciated and accepted!

1. The authors do almost nothing to situate the data reported with in a larger academic debate, and do not outline any theoretical perspective, specific hypothesis, or predictions. The most obvious perspective to bring to this manuscript is the developmental origins of health and disease (DOHaD) framework (which is sort of but not satisfactorily alluded to), although bio-cultural theory, dual inheritance theory, human behavioral ecology, evolutionary psychology, or some kind of network theory could also be compelling. Alternatively, an in depth ethnographic pharming describing the historical particulars (that is, the social as well as dietary significance of the food staff in the city and its sub-cultures) underpinning each of the common taboos would be interesting. Some intellectual framework needs to be established and justified for this to be publishable.

Response: Thank you very much for your valuable comments and suggestions dear. We have tried to refer articles that could explain the origin and background of food taboos and different theories associated with it as per your suggestion. Based on the scientific evidences that we have searched, we incorporated important points in the manuscript (lines 60 – 78)

2. The authors seemingly do not disentangle pregnancy food taboos from other kinds of food avoidances (visceral aversions, nausea, disgust, taboos not specific to pregnancy; avoidances due to public health guidelines; avoidances due to allergies or sensitivities or family preferences). It is not clear whether data were collected on whether pregnant people would usually eat a certain food and even wanted to eat it but did not because of a taboo or did not because it made them nauseous and there also happened to be a taboo about it. This issue needs to be better outlined in the methods; if data are available on other reasons for food avoidances, they should be reported and discussed; if data are not available, this should be discussed as a limitation and avenue for future research.

Response: Thank you for your invaluable and constructive comments dear! I will like to make it clear that we had clearly differentiated these terms in the very beginning of the study, and planned to assess the foods that are intentionally prohibited (taboo foods) during pregnancy for non-scientific reasons by the community in the study area. We have not collected a data on the other forms of food avoidances because we as public health nutrition professionals thought that we can easily make a public health intervention to address food taboos than the other forms of food avoidances. As per your suggestion, we have incorporated it in the limitation of the study (lines 317 – 325). 

3. No alternatives to the assumption that the taboos are problematic were presented. Yes, the pregnant people are missing out nutrients, but are they also avoiding pathogens, or preventing fetal overgrowth, or guarantying that nutrients go to support other household members? I would expect to see some thoughtful discussion and familiarity with the literature around these points. 

Response: We would like to appreciate for this valuable and important comment. It is true that food taboos would have positive and negative implications. As you mentioned it in your comment, people could avoid food related toxicity or poison through avoiding some food types. On the other hand, as we tried to focus in this study, food taboos would also have negative implications due to the risk of under nutrition. Therefore, we tried to address this issue that our main aim was to explore food taboos during pregnancy which could have negative implications (lines 79 – 85). 

4. The authors do not seem to look at the extent to which pregnant people are perhaps compensating for nutrients through experiencing cravings or following other dietary recommendations and thus filling in some of the nutrient gaps created by food avoidances.

• My colleagues and I found that there was evidence of pregnant people in Figi eating other foods with similar nutrient profiles to avoid foods (Mckerracher et al. 2016; see also Henrich and Henrich 2010).

Response: Thank you for raising a very crucial point dear! We tried to address the foods believed by the community to be bad during pregnancy for non-scientific reasons which can compromise the daily dietary intake of the mothers eventually affecting the health and nutritional status of the mothers and their growing baby. As you can see from table 2 of the manuscript, more than 3/4th (79.2%) of the pregnant women did not achieve the minimum dietary diversity score. 

5. Education is a major predictor of not adhering to food taboos, and the public health recommendation the authors make then is to improve antenatal education (example lines 271 – 272). I would suggest going to pre-conception and even to adolescence to really make a big difference. See the work by MacNab and Mukisa in South Africa, by Jackie Bay and colleagues in Newzealand and the Cooc Islands, by Mary Barker/Kathy Townsend and colleagues with the Each-B trial in Southampton UK and so on. Also check out the Lancet review on pre-conception interventions (Stephenson et al. 2018).

Response: We have made some modification based on your suggestion dear. We thought that short and long term interventions are required to address the issue (lines 326 – 332).

6. I would generally like to see a little more context on the overall health of the pregnant population in Mekelle in general and in the sample in particular. What are infant and maternal mortality like? Are there high rates of GDM? High rates of pre-eclampsia/ Hypertension? High rates of fetal growth restriction? Pre-term birth? Average completed family size? Food security? 

Response: Thank you very much for your valuable comment. We tried to search published scientific studies on these issues in the study area. However, we could not find studies regarding the magnitude of infant and maternal mortality rate, gestational diabetes mellitus, pre-eclamplsia/eclapsia, food security label, and fetal growth restrictions. We believe that these would be good areas of research that remained undiscovered. The average family size of the sample population were also 3.6 ≈ 4.

7. Please unpack the dietary diversity score. There should be clearly outlined in the main text. 

Response: Thank you for raising a valid point dear! We have narrated the women diet diversity individually as per your suggestion (lines 210 – 216). 

8. Lastly, the manuscript would benefit from a bit more spell and grammar checking. Hopefully, the editors can provide this or at least some financial assistance as the team is from a lower income country. I would also be happy to help with proofing the revision if I was sent a word doc. or Google doc. File.

Response: We tried to check the spelling and grammar of the whole manuscript as per your recommendation, and we would like to appreciate your willingness to help us in proofreading our manuscript and suggesting the editors to support us.

Comments raised by reviewer 2:

Thank you dear for your effort to make our manuscript better looking at it strictly and provision of constructive and invaluable comments. We accepted all of the comments as it is and here are the issues that need our response.

1. Line 30 – 31: the timeline should be specified as ‘’1st of January to 30th of June 2017. However, we wonder why this timeline of data collection is different from what is mentioned on line 87, where it was stated that ‘’the study was carried out from April to May 2017…’’ which one is correct? 

Response: Sorry for the mistake we made in writing the timeline dear! But the one in the abstract which says from 1st of January to 30th of June indicates the timeline starting from proposal development up to the final accomplishment of the study. Whereas, the timeline in line 87 shows the period of data collection. Up on your suggestion, I have revised it and write the one which indicates the data collection period from April to May 2017 in both cases.

2. Line 89 – 92: what is the source of information? Please, cite the source. What does ‘’2016/2017’’ mean? Does it refer for either of the years or for both?

Response: Thank you for raising a very important point dear! We brought the information from the Federal democratic republic of Ethiopian central statistics agency report 2017. I have put a reference in line 108.

3. Line 108: because your study clinics were randomly selected from the institutions with a ‘’better flow of pregnant women for ANC services’’, what does this imply to your findings? There could be the possibility of ‘’selection bias’’ because participants of your study were from clinics, which had a ‘’better flow’’ of pregnant women for ANC services. The profile (socio-demographic and economic attributes) of the participants of your study may differ from those who attended other health institutions which did not have a ‘’better flow’’, and the prevalence of food taboo and related misconceptions could be different. You may discuss it.

Response: Thank you for your nice question dear! In the beginning (proposal write-up) of the study, we made an assessment of the clinics for the services they were providing, and the flow of the pregnant women being served. The services delivered by the different clinics (with better and relatively poor flow of pregnant mothers) were similar in terms of their quality and cost. Therefore, we thought that the difference in the flow of the mothers might be emanated from popularity of the health providers working in the clinics and peer pressure (mothers may go to the clinics where their friends or relatives are being served). 

4. Line 109 and 112: if the allocation to each of the five health institutions was proportional, this implies the total number of pregnant women was known for each of these study clinics. How was the proportional allocation made? How was the sampling interval defined (N/n)? For a clarity purpose, you may submit a ‘’supplementary’ ’table or flow chart that shows the proportional allocation and its respective sampling interval. 

Response: Thank you for raising a very important point. Now I have attached the clarification as a supplementary material up-on your suggestion.

5. Line 137: on how many women and where was the pilot test implemented? Who were the actual participants of the pilot test study? 

Response: We conducted a pre-test rather than a pilot test dear. The questionnaire was pre-tested before the actual data collection on 5% (17) of the study participants in other similar clinics that were not included in the study. Some modifications were made based on the pretest results especially on the clarity of the questionnaire to the data collectors and respondents (lines 158 – 161).

6. Line 137 – 139: it is great that you did the translations and re-translations. Who did both translations to the local language and the re-translation back to English? Was it done by the same individual/s? You may briefly explain these points.

Response: Thank you for the noticing this nice point dear! The questionnaire was translated into Tigrigna (local language), and then back-translated to English by two individuals independently for ensuring the consistency of concepts (line 163).

7. We strongly suggest using appropriate statistical analysis to identify the factors associated with food taboo, which has a prevalence of 11.5%. Authors should use the data to its maximum potential and may have ethical aspect from statistical point of view. We suggest to the authors two important points (based on their result from table 4).

• 1st, they should combine the levels categories of some independent variables in to broader categories to avoid the problem of convergence.

• 2nd, they should test for presence of multi-collinearity, which could bias their measure of association like the odds ratio or prevalence ratio. This is not a must to do, but we encourage you to do it.

• If the authors decide to run a model to identify the factors associated with food taboo, the objective and methods part of the study need to be modified accordingly.

Response: A very nice suggestion dear! In the very beginning, our difficulty was to merge the variables. Now, we have run a new analysis by merging some of the variable in to broader categories up on your suggestion. We assessed the socio-demographic factors associated with food taboo practice among pregnant women using bivariate and multivariable logistic regression analysis. Accordingly, we made necessary modifications in the objective (lines 22 &23), methods (lines 168 – 185), result (lines 238 – 246), and discussion (lines 309 – 316) of the study.

8. Line 160: age of participants was reported in mean and standard deviation. Was it normally distributed? If not, you need to report the median and inter-quartile range. This works for all continuous variables you have (like family size).

Response: We have checked all of the continuous variables for normality using Shapiro Wilk’s test and they were found to be appropriately normally distributed (p> 0.05) (lines 170 – 173). 

9. Line 180: report the 95% CI of the prevalence.

Response: Oh! Sorry for missing this important point dear! Thank you for seriously looking at everything in our manuscript dear! Now I have incorporated it (line 253)

10. Line 249 – 253: this lacks critical interpretation of the results to create awareness on the potential harms of consuming honey on the health of their newborns. You should also critically discuss the various negative aspects of honey use. You need to site more relevant articles in this field, which can help you to improve the respective discussion.

a. Please site more important articles, you should at least site the work of Ajibola et al. (doi: 10.1186/1743-7075-9-61). 

b. Natural honey can be contaminated by C.botulinum, which is fatal and definitely, should not be given to children less than one year. Check CDC’s official webpage (https://www.cdc.gov/botulism/)

Response: Thank you for raising a very important and educative point dear! It is obvious that consumption of honey by children under one year has many negative implications related with the toxicity effect of C.botulinum. However, this study mainly focused on the consumption of natural honey by pregnant women and its health and nutrition implications to the mother and her growing fetus before she give birth. Thus, we could not find any evidence which suggests to prevent pregnant mothers from consuming natural honey because of consequences to them and their fetus. The articles you suggested us as well are about consumption of natural honey after delivery and before one year of age.

---

## [Decision Letter · Decision Letter 1]

10 Jun 2020

PONE-D-20-06088R1

Food taboos and related misperceptions during pregnancy in Mekelle city, Tigray, Northern Ethiopia

PLOS ONE

Dear Dr. Tela,

Thank you for submitting your manuscript to PLOS ONE. After careful consideration, we feel that it has merit but does not fully meet PLOS ONE’s publication criteria as it currently stands. Therefore, we invite you to submit a revised version of the manuscript that addresses the points raised during the review process.

SPECIFIC ACADEMIC EDITOR COMMENTS: The same reviewers handled your revised manuscript. There were still major issues found in your study. One of the most important issues that needs to be addressed is related to the development of a testable, directional hypothesis - reviewer 1 offers suggestions for models to run to begin examining cause or effect relationships. Reviewer 2 has important comments related to the statistics and English grammar.

We look forward to receiving your revised manuscript.

Kind regards,

Frank T. Spradley

Academic Editor

PLOS ONE

Reviewers' comments:

Reviewer's Responses to Questions

**Comments to the Author**

1. If the authors have adequately addressed your comments raised in a previous round of review and you feel that this manuscript is now acceptable for publication, you may indicate that here to bypass the “Comments to the Author” section, enter your conflict of interest statement in the “Confidential to Editor” section, and submit your "Accept" recommendation.

Reviewer #1: (No Response)

Reviewer #2: (No Response)

2. Is the manuscript technically sound, and do the data support the conclusions?

Reviewer #1: Yes

Reviewer #2: Yes

3. Has the statistical analysis been performed appropriately and rigorously? 

Reviewer #1: Yes

Reviewer #2: Yes

4. Have the authors made all data underlying the findings in their manuscript fully available?

Reviewer #1: Yes

Reviewer #2: (No Response)

5. Is the manuscript presented in an intelligible fashion and written in standard English?

Reviewer #1: Yes

Reviewer #2: No

6. Review Comments to the Author

Reviewer #1: As per my previous review of this paper, I believe you report important data with obvious public health and social implications. Further, you have addressed the majority of my specific concerns regarding the need to provide a little more ethnographic context, to unpack some of your measures (e.g. dietary diversity), and the possible benefits of food taboos.

However, the paper still lacks any sort of clear theoretical frame. Maybe this is down to a disciplinary difference? I come from anthropology, and USING AND DEVELOPING THEORY IS ESSENTIAL TO ACADEMIC PUBLICATION. What debate(s) does this study move forward? You claim that you addressed this issue in lines 60-78, but this is just a shallow reading of some evidence suggesting that food taboos may sometimes have benefits, and are variable from society to society. It doesn't offer any suggestion as to WHY taboos vary among populations, and what that might mean for the study population (in terms of their diets, their current health, their future health, the health of the next generation, the wider environmental context in Mekelle). I don't know, man. I would still call this not even close to ready for publication until some deep thought has been put into theory, hypotheses, and predictions. But, on the other hand, I work quite a bit with public health researchers and health scientists and they're not as worried about theory as anthropologists (or sociologists or biologists or psychologists - the home disciplines of my main collaborators) tend to be. So, I guess it'll be up to the editor whether this cuts the mustard?

One other point that flows from the first (i.e., the lack of theoretical clarity and predictions): From my read of the data (not from your non-presentation of theory), you seem like you might be interested in food taboos as a possible proximate CAUSE of low dietary diversity during pregnancy. Why don't you test this? Would be a super simple model to run. Do women who report adhering to food taboos have lower dietary diversity scores/ are less likely to meet the MDD cut-off? If so, is this independent of education, control over financial resources? If not, what's the point of the paper? I mean, it'd still be interesting data - you'd just need to figure out a different angle (like, woah, lack of access to control over household resources, for example, is actually a more important driver of low MDD than taboos).

Lastly, I do not want to go through line by line of a pdf writing suggestions for how to improve the clarity of the language given that I think it needs another round of substantial revision. But, if the editor decides to accept with minor revisions and no additional review, I ask you (the authors) to please send me your text in a word doc and I will happily spend an hour cleaning up the language (no charge or expectation of credit, obviously - just paying academic support forward!). I'm at mckerrl@mcmaster.ca.

Reviewer #2: 1. Page7, Ln133-134: please, put the numerator and denominator in bracket and multiply it by the “sample size”

2. age9, Ln180: multi-co linearity should be written as “multicolinearity”

3. Page9, Ln181: It is uncommon to use the standard errors to check multicolinearity, VIF is common. Anyway, put a reference for the “standard error of > 2” criterion.

4. Page 11, Ln201 (Table 1): please don’t underline any word in scientific words. Don’t use contractions in research works. So, avoid the contractions like “No” and “≥5 childbirths” and write it like “number” and “≥5 childbirths”. Please, check for such other things throughout the manuscript.

7. PLOS authors have the option to publish the peer review history of their article (what does this mean?). If published, this will include your full peer review and any attached files.

Reviewer #1: Yes: Luseadra McKerracher

Reviewer #2: Yes: Semaw Ferede Abera

---

## [Author Response · Author response to Decision Letter 1]

19 Aug 2020

Dear our editor and reviewers! Thank you for your relentless effort to improve the quality of our manuscript. All of your comments and suggestions are really appreciated and accepted!

Reviewer #1: 

1. As per my previous review of this paper, I believe you report important data with obvious public health and social implications. Further, you have addressed the majority of my specific concerns regarding the need to provide a little more ethnographic context, to unpack some of your measures (e.g. dietary diversity), and the possible benefits of food taboos.

However, the paper still lacks any sort of clear theoretical frame. Maybe this is down to a disciplinary difference? I come from anthropology, and USING AND DEVELOPING THEORY IS ESSENTIAL TO ACADEMIC PUBLICATION. What debate(s) does this study move forward? You claim that you addressed this issue in lines 60-78, but this is just a shallow reading of some evidence suggesting that food taboos may sometimes have benefits, and are variable from society to society. It doesn't offer any suggestion as to WHY taboos vary among populations, and what that might mean for the study population (in terms of their diets, their current health, their future health, the health of the next generation, the wider environmental context in Mekelle). I don't know, man. I would still call this not even close to ready for publication until some deep thought has been put into theory, hypotheses, and predictions. But, on the other hand, I work quite a bit with public health researchers and health scientists and they're not as worried about theory as anthropologists (or sociologists or biologists or psychologists - the home disciplines of my main collaborators) tend to be. So, I guess it'll be up to the editor whether this cuts the mustard?

Response: Many thanks for your constrictive as well as valuable comments. Based on the comments given we tried to clarify why taboos vary from population to population mainly taking into consideration the beliefs and practices of the study area. We also tried to address what would be the possible fate of practicing food taboo to the people in the study area mainly pregnant mothers, its implication to the next generation, and we also hypothesized what would happen if proper intervention is not done. Furthermore, we tried to describe the fate of globalization towards food taboo mainly in the new generation. We have addressed the points in the introduction section of our manuscript in lines; 64-71; 76-100; and 112-116. We also inserted our references for the evidences in the reference list numbers; 7-11 and 14-18. 

2. You seem like you might be interested in food taboos as a possible proximate CAUSE of low dietary diversity during pregnancy. Why don't you test this? Would be a super simple model to run. Do women who report adhering to food taboos have lower dietary diversity scores/ are less likely to meet the MDD cut-off? If so, is this independent of education, control over financial resources? If not, what's the point of the paper? I mean, it'd still be interesting data - you'd just need to figure out a different angle (like, woah, lack of access to control over household resources, for example, is actually a more important driver of low MDD than taboos).

Response: We would like to appreciate your constructive comment. Though it was out of our research objective taking into consideration the importance of your constructive comment, we have analyzed the association between food taboos and MDDS. Based the bivariate analysis, we could observe that the number of pregnant mothers who reported experience of food taboo and had low MDDS were higher than mothers who reported no food taboo and had low MDDS. However, the finding was not statistically significant (31 (11.8%), 7 (10.8%), p value 0.703, respectively. We believe that we could see significant difference among the mothers if our sample size was big. Farther research with bigger sample size to assess the relationship between MDDS and food taboos during pregnancy shall be conducted. 

Reviewer #2: 

1. Page7, Ln133-134: please, put the numerator and denominator in bracket and multiply it by the “sample size”

2. age9, Ln180: multi-co linearity should be written as “multicolinearity”

3. Page9, Ln181: It is uncommon to use the standard errors to check multicolinearity, VIF is common. Anyway, put a reference for the “standard error of > 2” criterion.

4. Page 11, Ln201 (Table 1): please don’t underline any word in scientific words. Don’t use contractions in research works. So, avoid the contractions like “No” and “≥5 childbirths” and write it like “number” and “≥5 childbirths”. Please, check for such other things throughout the manuscript.

Response: Many thanks for your valuable comments. We have changed the methods of assessing multicolinearity in our model using the commonly used techniques, the variance inflation factor (VIF). Accordingly all independent variables had VIF less that 5 and no variables were excluded from the analyses. We explained this in the data analysis and management section of our manuscript between lines 213-215. We also inserted our reference/evidence for this justification in the reference list number 13. Moreover, we have addressed all the constructive comments given including the grammar (language) checks and other comments in the manuscript.

---

## [Decision Letter · Decision Letter 2]

25 Aug 2020

PONE-D-20-06088R2

Food taboos and related misperceptions during pregnancy in Mekelle city, Tigray, Northern Ethiopia

PLOS ONE

Dear Dr. Tela,

Thank you for submitting your manuscript to PLOS ONE. After careful consideration, we feel that it has merit but does not fully meet PLOS ONE’s publication criteria as it currently stands. Therefore, we invite you to submit a revised version of the manuscript that addresses the points raised during the review process.

ACADEMIC EDITOR COMMENTS: There are some remaining recommendations from both reviewers that are required to be included in your revised manuscript.

We look forward to receiving your revised manuscript.

Kind regards,

Frank T. Spradley

Academic Editor

PLOS ONE

Reviewers' comments:

Reviewer's Responses to Questions

**Comments to the Author**

1. If the authors have adequately addressed your comments raised in a previous round of review and you feel that this manuscript is now acceptable for publication, you may indicate that here to bypass the “Comments to the Author” section, enter your conflict of interest statement in the “Confidential to Editor” section, and submit your "Accept" recommendation.

Reviewer #1: (No Response)

Reviewer #2: All comments have been addressed

2. Is the manuscript technically sound, and do the data support the conclusions?

Reviewer #1: Yes

Reviewer #2: (No Response)

3. Has the statistical analysis been performed appropriately and rigorously? 

Reviewer #1: Yes

Reviewer #2: Yes

4. Have the authors made all data underlying the findings in their manuscript fully available?

Reviewer #1: Yes

Reviewer #2: Yes

5. Is the manuscript presented in an intelligible fashion and written in standard English?

Reviewer #1: No

Reviewer #2: Yes

6. Review Comments to the Author

Reviewer #1: Dear Authors,

I think generally you've addressed my most important comments from the previous draft. In particular, you've now framed the background in a way that foregrounds the questions: "why might there be food taboos during pregnancy?" and "How might those taboos affect maternal and child health?" . Furthermore, you have run the additional analysis that I suggested you respect to whether diet diversity score is related to reporting following taboos.

I have 2 remaining recommendations though:

1) Actually report the results of the analysis testing for an association between diet quality and food aversions, and then briefly explain in the discussion the lack of relationship (could be insufficient power to detect an effect, could be that socio-economic and/or political factors are much more important drivers of variation in diet quality).

2) Get me or someone else who is not part of your team to go through a word doc version of the text and edit/ proofread it. There are too many grammatical, word choice, and phrasing errors to fix by just noting them by line, otherwise I'd do it now with the journal-generated PDF. There are several places where these errors make the argument difficult to parse, so it'd be a much more solid paper with a little editorial TLC.

If you do these 2 things, In my view, it's acceptable/ ready for publication. I don't need to see an additional revision to greenlight it! Well done.

Reviewer #2: I congratulate the authors for their extensive review and resubmission! All my previous comments are addressed. However, the authors need to make two minor revisions. The fullstops on Page5, Line 89 and Page5, Line 100 are in "red" color and this is not allowed; change the red colors to black. Please, make sure that there are no such errors throughout your document, including in your supplementary files.

7. PLOS authors have the option to publish the peer review history of their article (what does this mean?). If published, this will include your full peer review and any attached files.

Reviewer #1: **Yes: **Luseadra McKerracher

Reviewer #2: **Yes: **Semaw Ferede Abera

---

## [Author Response · Author response to Decision Letter 2]

6 Sep 2020

Response to reviewers

Reviewer #1: Dear Authors,

I think generally you've addressed my most important comments from the previous draft. In particular, you've now framed the background in a way that foregrounds the questions: "why might there be food taboos during pregnancy?" and "How might those taboos affect maternal and child health?". Furthermore, you have run the additional analysis that I suggested you respect to whether diet diversity score is related to reporting following taboos.

I have 2 remaining recommendations though:

1) Actually report the results of the analysis testing for an association between diet quality and food aversions, and then briefly explain in the discussion the lack of relationship (could be insufficient power to detect an effect, could be that socio-economic and/or political factors are much more important drivers of variation in diet quality).

Thank you very much for your valuable and constructive comments

We have incorporated the result of the analysis on the relation between MDDS and practice of food taboos. We put the results on the results section of our manuscript between lines 300 and 302. And on the discussion section from lines 431 up to 439.

2) Get me or someone else who is not part of your team to go through a word doc version of the text and edit/ proofread it. There are too many grammatical, word choice, and phrasing errors to fix by just noting them by line, otherwise I'd do it now with the journal-generated PDF. There are several places where these errors make the argument difficult to parse, so it'd be a much more solid paper with a little editorial TLC.

If you do these 2 things, In my view, it's acceptable/ ready for publication. I don't need to see an additional revision to greenlight it! Well done.

Thank you very much for your valuable comments.

We would love to greatly appreciate for the commitment of our first reviewer Dr. Luseadra McKerracher for her precious time invested to do the language and scientific edit of our manuscript. Now the manuscript is edited for language and scientific clarity. We would suggest the journal to recognize her commitment in any means if she is interested on it.

Reviewer #2: I congratulate the authors for their extensive review and resubmission! All my previous comments are addressed. However, the authors need to make two minor revisions. The fullstops on Page5, Line 89 and Page5, Line 100 are in "red" color and this is not allowed; change the red colors to black. Please, make sure that there are no such errors throughout your document, including in your supplementary files.

We would like to thank you for your valuable comments.

We have changed the color to black. We also checked and corrected the entire document and all the supplementary tables.

---

## [Editor Report · Decision Letter 3]

7 Sep 2020

Food taboos and related misperceptions during pregnancy in Mekelle city, Tigray, Northern Ethiopia

PONE-D-20-06088R3

Dear Dr. Tela,

We’re pleased to inform you that your manuscript has been judged scientifically suitable for publication and will be formally accepted for publication once it meets all outstanding technical requirements.

Kind regards,

Frank T. Spradley

Academic Editor

PLOS ONE

---

## [Editor Report · Acceptance letter]

10 Sep 2020

PONE-D-20-06088R3 

Food taboos and related misperceptions during pregnancy in Mekelle city, Tigray, Northern Ethiopia 

Dear Dr. Tela:

I'm pleased to inform you that your manuscript has been deemed suitable for publication in PLOS ONE. Congratulations! Your manuscript is now with our production department. 

Kind regards, 

on behalf of

Dr. Frank T. Spradley 

Academic Editor

PLOS ONE